# Inhibition of Tumor Growth by Dietary Indole-3-Carbinol in a Prostate Cancer Xenograft Model May Be Associated with Disrupted Gut Microbial Interactions

**DOI:** 10.3390/nu11020467

**Published:** 2019-02-22

**Authors:** Yanbei Wu, Robert W. Li, Haiqiu Huang, Arnetta Fletcher, Lu Yu, Quynhchi Pham, Liangli Yu, Qiang He, Thomas T. Y. Wang

**Affiliations:** 1College of Light Industry, Textile and Food Engineering, Sichuan University, Chengdu 610065, China; yanbeiwu@hotmail.com; 2Department of Nutrition and Food Science, University of Maryland, College Park, MD 20742, USA; afletche@shepherd.edu (A.F.); yulu0640514@gmail.com (L.Y.); lyu5@umd.edu (L.Y.); 3Diet, Genomics, and Immunology Laboratory, Beltsville Human Nutrition Research Center, USDA-ARS, Beltsville, MD 20705, USA; tennisqiu@gmail.com (H.H.); quynhchi.pham@ars.usda.gov (Q.P.); 4Animal Parasitic Diseases Laboratory, USDA-ARS, Beltsville, MD 20705, USA; robert.li@ars.usda.gov; 5Department of Family and Consumer Sciences, Shepherd University, Shepherdstown, WV 25443, USA

**Keywords:** indole-3-carbinol, prostate cancer, gut microbiota, microbial interactions, co-occurrence network

## Abstract

Accumulated evidence suggests that the cruciferous vegetables-derived compound indole-3-carbinol (I3C) may protect against prostate cancer, but the precise mechanisms underlying its action remain unclear. This study aimed to verify the hypothesis that the beneficial effect of dietary I3C may be due to its modulatory effect on the gut microbiome of mice. Athymic nude mice (5–7 weeks old, male, Balb c/c nu/nu) with established tumor xenografts were fed a basal diet (AIN-93) with or without 1 µmoles I3C/g for 9 weeks. The effects of dietary I3C on gut microbial composition and microbial species interactions were then examined by 16s rRNA gene-based sequencing and co-occurrence network analysis. I3C supplementation significantly inhibited tumor growth (*p* < 0.0001) and altered the structure of gut microbiome. The abundance of the phylum Deferribacteres, more specifically, *Mucispirillum schaedleri*, was significantly increased by dietary I3C. Additionally, I3C consumption also changed gut microbial co-occurrence patterns. One of the network modules in the control group, consisting of seven bacteria in family S-27, was positively correlated with tumor size (*p* < 0.009). Moreover, dietary I3C disrupted microbial interactions and altered this association between specific microbial network and tumor development. Our results unraveled complex relationships among I3C ingestion, gut microbiota, and prostate tumor development and may provide a novel insight into the mechanism for the chemopreventive effect of dietary I3C on prostate cancer.

## 1. Introduction

Prostate cancer was the third leading cause of cancer-related death worldwide in 2018 [1]. It has a long incubation period before clinical manifestation and is typically diagnosed in men, 50 years old or older [2]. Genetic as well as environmental factors, especially dietary patterns, have been reported to be involved in prostate cancer development [3,4]. A long latency and strong environmental conditions make prostate cancer an ideal target for nutritional intervention. Furthermore, a number of diet-derived bioactive compounds, such as lycopene, vitamin D, selenium, and indole-3-carbinol, have been reported to exert protective effects on prostate cancer development [5]. However, the precise mechanisms by which diet or diet-derived compounds protect against prostate cancer remain unclear.

Cruciferous vegetables, including broccoli, cabbage, and cauliflower, are inversely associated with prostate cancer incident. It has been reported that consuming three or more serving of cruciferous vegetables per week could reduce the risk of developing prostate cancer by 40% [6]. Due to such potential benefits, the identification of natural, anti-carcinogenic bioactive compounds found in cruciferous vegetables has gained widespread attention. Indole-3-carbinol (I3C) is derived from naturally occurring glucosinolates in cruciferous vegetables. It is rapidly converted into a range of metabolites under the acidic stomach environment, among which 3,3-diindolylmethane (DIM) is the most prominent compound [7]. These indoles are known for their chemopreventive effects on hormone-dependent cancers such as prostate cancer [8]. It has been shown that I3C/DIM inhibit proliferation, migration, and invasion of prostate cancer cells in vitro by targeting a wide spectrum of signal pathways governing hormonal homeostasis, cell cycle progression, apoptosis, and DNA repair [9,10,11]. Several nuclear transcription factor-mediated pathways related to these biological processes have been reported to be modulated by I3C/DIM [12]. However, the precise mechanisms by which I3C or DIM may work to protect against prostate cancer remain unclear and warrant further elucidation.

Recently, emerging evidence supports work on the trillions of bacteria residing in the human gastrointestinal (GI) tract form a complex ecological community, commonly known as the gut microbiota [13]. Published literature also supports that the gut microbiota considerably impacts on host health by modulating immune response, intestinal homeostasis, nutrients processing, energy harvesting, and resistance to pathogens [14,15,16]. Many factors, including dietary, have been implicated in the establishment and development of gut microbiota in individuals [17,18]. Moreover, alteration of the gut microbiota may also arise in diverse clinical situation such as obesity, inflammatory bowel disease, and cancer [19]. A recent study by Golombos et al. [20] indicated that biologically significant differences exist in the gut microbial composition of men with prostate cancer compared to benign controls. Higher relative abundance of *Bacteriodes massiliensis* was seen in the prostate cancer cases compared to the controls. In contrast, *Faecalibacterium prausnitzii* and *Eubacterium rectalie* had higher relative abundance among the control subjects. In addition, a study by Frugé et al. reported that Gleason score was positively associated with the phylum Deferribacteres (*p* < 0.032) and several Proteobacteria taxa. Gleason score was also positively associated with Clostridium (*p* < 0.005) and inversely associated with *Blautia* (*p* < 0.049) [21]. These studies suggest a possible link between the gut microbiota and development of prostate cancer. More importantly, a critical modulatory roles of dietary bioactive compounds on the gut microbiota has emerged as an important factor in the etiology, prevention, and therapy of various diseases, including cancer [13,22]. However, literature related to the effects of diet or diet-derived compounds such as I3C on the gut microbiota are scarce. Thus, it is unclear whether a relationship between dietary ingestion of I3C, gut microbiota, and prostate cancer development exists.

The present study seeks to address the deficiency in the literature and test the hypothesis that the prostate cancer protective effects of I3C against prostate cancer may be associated with modulation of the gut microbiota. In this study, the impact of I3C on gut microbiota composition was investigated in a mouse prostate cancer xenograft model using a metagenomics approach. Furthermore, we sought to delineate the complex interactions among I3C, gut microbiome, and prostate cancer development using co-occurrence network analysis.

## 2. Materials and Methods

### 2.1. Animals and Diet

Male athymic nude mice (BALB/c nu/nu, 20–22 g, 6–8 weeks old) were purchased from Charles River (Frederick, MD, USA) and were individually housed in filter-top cages at the Beltsville Human Nutrition Center’s animal facility under 12-h light cycle. An acclimation period of 1 week prior to treatment feedings enabled mice to adapt to their environment. There were nine animals in each group. Animals were then randomized into two experimental feeding groups: (i) control diet, (ii) control diet with 1 µmol I3C/g diets. Animals were fed diet and filtered water ad libitum. Food consumption and body weights were recorded weekly. This study was approved by the USDA Animal Research Advisory Committee, Beltsville Area Animal Care and Use Committee protocol #12-030.

### 2.2. Tumor Xenograft Model

LNCaP (ATCC CRL-1740), an androgen-dependent human prostate carcinoma cell line, was grown in Roswell Park Memorial Institute 1640 medium (RPMI 1640) supplemented with 10% fetal bovine serum (FBS), 100 U/mL penicillin, and 100 µg/mL streptomycin purchased from Invitrogen (Carlsbad, CA, USA). Cultures were incubated in an atmosphere composed of 5% CO_2_ at 37 °C.

After a two-week dietary intervention, all mice were inoculated with LNCaP cell suspension via subcutaneous injection into the right and left flanks. Cell suspension consisted of LNCaP cells at a density of 2 × 10^6^ with 50 µL of phosphate-buffered saline (PBS) and an equal volume of Matrigel (BD Biosciences, Mansfield, MA, USA). Mice remained on their respective diets for 7 weeks following inoculation. Injection sites and tumor volume (cm^3^) were monitored and measured weekly for palpable tumor growth and biological efficacy: (cm^3^) = 0.523 × [length (cm) × width^2^ (cm^2^)] [23,24].

### 2.3. Fecal DNA Extraction and Analysis

The mouse fecal pellet was homogenized with Precellys (Bertin Technologies, Saint-Aubin France) at 7500 rpm for 1 min. Then fecal DNA was extracted using a QIAamp DNA Stool Mini Kit from Qiagen (Valencia, CA, USA) following manufacturer’s protocol with modification [25]. DNA was eluted from the column with 100 μL nuclease-free water. The concentration of DNA elution was determined by its absorbance at 260 nm, followed by serial dilutions to the final concentration of 10 ng/μL. Bacterial groups were quantified by quantitative PCR using Applied Biosystems 7900T Real-Time PCR System (Applied Biosystems™, Forest City, CA, USA) and the group-specific primers are shown in Table 1. Real-time PCR was performed on Applied Biosystems 7900T Real-Time PCR System using 10 μL SYBR^®^ Green Real-Time PCR Master Mix (Applied Biosystems™, Forest City, CA, USA), 0.25 μL 500 nM custom-made oligo primers, 4.5 μL water, and 5 μL 10 ng/μL DNA [26,27]. The relative expression levels of bacteria in fecal samples were calculated from threshold cycle values.

### 2.4. Cecal DNA Extraction, Amplicon Library Construction and Sequencing

Cecal samples were obtained at the time of sacrifice and quick-frozen in liquid nitrogen, then stored at −80 °C until needed. Total DNA was extracted from cecum contents using QIAamp Fast DNA Stool Mini Kit (Qiagen, Valencia, CA, USA) with some modifications. The hypervariable regions V3-V4 of the 16S rRNA gene were amplified from 20 ng of total DNA with a sample-specific barcode using 2.5 units of AccuPrimeTM Taq DNA Polymerase (Invitrogen, Carlsbad, CA, USA) in a 50 µL reaction buffer containing 200 nM primers, 200 nM dNTP, 60 mM Tris-SO_4_, 18 mM (NH_4_)_2_SO_4_, 2.0 mM MgSO_4_, 1% glycerol, and 100 ng/µL bovine serum albumin (New England Biolabs, Beverly, MA, USA). PCR was performed under the following cycling conditions: initial denaturing at 95 °C for 2 min followed by 20 cycles of 95 °C for 30 s, 60 °C for 30 s, and 72 °C for 60 s. Amplicons were purified using Agencourt AMPure XP bead kits (Beckman Coulter, Fullerton, CA, USA) and quantified using a High Sensitivity DNA Kit (Agilent, Santa Clara, CA, USA). The purified amplicons from each sample were pooled in equal molar ratios and further spiked with approximately 25% of whole-genome shotgun libraries prepared using an Illumina TruSeq DNA sample prep kit (Illumina, San Diego, CA, USA) with a compatible adaptor barcode to enhance sequence diversity during the first few cycles of sequencing for better cluster differentiation. The final concentration of the library pool was determined using a Bio-Analyzer high-sensitivity DNA chip kit (Agilent, USA). The library pool was sequenced using an Illumina MiSeq Reagent Kit and Illumina MiSeq sequencer [28,29].

### 2.5. Sequence-Based Microbiome Analysis

The sequence data were preprocessed using MiSeq Control Software (MCS) version 2.4.1 (Illumina, San Diego, CA, USA). Raw sequences were first analyzed using FastQC version 0.11.2 (Babraham Institute, Cambridge, UK) to check basic statistics, such as %GC, per base quality score distribution, and sequences flagged as poor quality. The four maximally degenerate bases (“NNNN”) at the 5′ end of the read pair, which were designed to maximize the diversity during the sequencing run of first four bases for better identification of unique clusters and improve base-calling accuracy, were then removed. The presence of forward and reverse PCR primers at the 5′ and 3′ ends of each sequence read was scanned and the reads without primers sequences were discarded. The chimeric reads were also removed from the dataset. The processed pair-end reads were then merged using PandaSeq version 2.8 (University of Waterloo, Waterloo, ON, Canada) to generate representative complete nucleotide sequences (contigs) using default parameters. The overlapping regions of the pair-end read were first aligned and scored, and reads with low score alignments and high rate of mismatches were discarded.

QIIME pipeline (version 1.9.1, University of Colorado Boulder, Boulder, CO, USA) was used to analyze the 16S rRNA gene sequences. The sequences with ≥97% identity were binned into operation taxonomic unit (OTU) according to a “closed reference” protocol. GreenGene database (version 13.8) was used for taxonomy assignment (greengenes.lbl.gov). PyNAST (version 1.2.2, University of Pennsylvania School of Medicine, Philadelphia, PA, USA) was used for sequence alignment. The microbial diversity in the murine intestine was analyzed using QIIME pipeline based on the OTU table. OTU relative abundance values were then analyzed using the LEfSe algorithm to identify taxa that displayed significant differences between two biological conditions. -The LEfSE uses the linear discriminant analysis (LDA) to estimate the - size of each differentially abundant feature.

### 2.6. Network Construction and Analysis

The network analysis (control and I3C treatment) was performed by Random-Matrix theory (RMT)-based pipeline, as described by Zhou et al. [30]. The network construction was based on the OTU abundance table. The OTUs that were detected in less than 50% of all samples were excluded for further analysis. A similarity matrix, which measures the degree of concordance between the abundance profiles of individual OTUs across different samples [30], was then obtained by using Pearson correlation analysis of the abundance data [31]. The threshold values used in this study were 0.87 for the control group and 0.92 for I3C group. The fast-greedy modularity optimization procedure was used for module separation. The within-module degree (Zi) and among-module connectivity (Pi) were then calculated and plotted to generate a scatter plot for each network to gain insights into the topological roles of individual nodes in the network. The Olesen classification approach was used to define node topological roles [32]. A Mantel test was performed to measure the relationship of the network topology and physiological traits by calculating OTU significance and node connectivity, as described by Zhou et al. [30]. Finally, the network was visualized using Cytoscape version 3.1.0 (National Institute of General Medical Sciences, Bethesda, MD, USA).

## 3. Results

### 3.1. The Effect of I3C on the Cecal Microbiota in Mice

To investigate specific effects of I3C on gut microbiota, cecum contents were collected for 16S RNA sequencing analysis [28]. The species richness from the result of the sample was estimated by the construction of rarefaction curves. The curves (Appendix A) grew rapidly at first and then began to flatten as fewer new species were being found per sample, supporting that the sequencing depth in our study was adequate [33]. Approximately 767 OTUs were detected in the microbial community of mice injected with LNCaP cells fed with control and I3C-containing or supplemented diets, respectively. Common microbial alpha-diversity indices, such as Chao 1, PD_whole_tree, Shannon and Simpson, were evaluated. The results indicated that the I3C diet did not significantly alter the species-level microbial diversity in the cecal microbiota of mice (Appendix A).

Further analysis at the phylum level revealed that the most abundant phylum in the gut microbiota (Figure 1A) was Firmicutes, which account for about 52.8% of all sequences, followed by Bacteroidetes (44.8%) and Deferribacteres (2.26%). I3C in the diet significantly increased the abundance of Deferribacteres (*p* = 0.01), which was considered as a biomarker (LDA score = 3.8) of the I3C group. While the reduction of Firmicutes and the increase of Bacteroidetes were observed, neither reached a statistically significant level in cecal samples (Figure 1A).

At species level, a total of 14 OTUs were significantly altered in relative abundance based on Wilcoxon non-parametric *t*-test corrected for multiple hypothesis testing (log_10_LDA > 2.0, Table 2), among which 13 OTUs belonged to the phylum Firmicutes. Ten of them increased in I3C-fed animals. The abundance of OTU#343923, in particular, was 55-fold higher in the I3C group than that in the control group. OTU#1136443, which belongs to the phylum Deferribacteres, represented a higher ratio in the microbial community than other changed OTUs. The abundance of this specie was increased from 2.26% to 3.5% in the I3C diet group.

The microbial cladogram (as shown in Figure 1B) indicates that the gut microbiome was most significantly altered in Deferribacteres, ranging from phylum to species level. At species level, the relative abundance of *Mucispirillum schaedleri* was enriched (Figure 1C) in mice fed the I3C diet.

### 3.2. The Effect of I3C on Microbial Co-Occurrence Network in Mice

In order to investigate whether I3C affects interactions between the intestinal bacteria, global microbial co-occurrence networks were constructed using Phylogenetic Molecular Ecological Network pipeline, an online tool based on a Random Matrix Theory (RMT)-based method [28,29]. The topological properties of microbial co-occurrence networks inferred in the control (C) and I3C group (I) are listed in Appendix A. The general properties of these two networks look similar. However, the composition and structure of the network in each group are substantially different (Figure 2). There were eight interaction modules identified in the control and I3C treatment group, respectively. While the individual module in the two groups consisted of different nodes (size) and shapes (connections), most of the modules in each network appeared to be unique. There were no exact matches for any modules in the two global networks. However, some modules in the control group shared the same member with modules in the I3C group. For example, 15 members were shared between module 1 in the control group and module 6 in the I3C group. All of members in module 8 of the control group were also contained in module 7 of the I3C group, suggesting that these two modules may be functional equivalents in the microbial community.

The microbial network, consisting of the species (nodes) by pair interactions (link), holistically presents various biological interactions in an ecosystem. Each node plays a distinct role in the network. The topological roles of the nodes identified in these two networks were further visualized as scatter plots (Figure 3) using two parameters: within-module connectivity (Zi) and among-module connectivity (Pi). The majority of the nodes (>98%) were identified as peripherals (Zi < 2.5, Pi < 0.62), indicating few links outside of their modules. Only three nodes were identified as module hubs in the two networks. These nodes were highly connected to other nodes within their own module (Zi > 2.5, Pi < 0.62). Interestingly, the unique module hub (OTU 452823) in the I3C network belonged to Lactobacillales, while the two module hubs (OTU 349537 and OTU 273778) in the control group were assigned to Clostridiales. Three nodes (control group: OTU 276006, I3C group: OTU 275474 and OTU 349036) were defined as connecters in the network. Although the connecters all belonged to Clostridiales, they are totally different at species level. In addition, no network hubs were observed in these two global-networks.

### 3.3. Association between I3C-Altered Microbial Interactions and Tumor Growth

In addition to interactions among microbes within the community, the relationships between module-based eigengenes and physiological traits can be used to detect the module’s response to physiological changes. These data are critical to predict the functional role of microbiota in disease processes. In our LNCaP xenograft model, tumor growth was significantly attenuated by the consumption of I3C (*p* < 0.0001, Figure 4A). To elucidate possible correlations between corresponding module eigengenes and tumor size, OTU significance and nodes’ topological indices were calculated by mental tests [31].

As shown in Figure 4B, module 8 in the control network showed a significant (*p* < 0.009) positive correlation with tumor growth. All nodes in this module were assigned to S24-7 family (Figure 4C). More importantly, this association was perturbed by consumption of the I3C diet. No module in the network of the I3C group displayed any association with tumor growth.

### 3.4. The Effect of I3C on the Fecal Microbiota in Mice

For comparison, microbial communities were also analyzed in feces using a quantitative PCR-based assay. The selected primers (Table 1), which target major phyla of fecal microorganisms, were used to quantify their abundances. As shown in Figure 5, the I3C diet had no effect on the abundance of total bacteria as well as Bacteroidetes (*p* > 0.05). However, I3C appeared to significantly reduce the abundance of Firmicutes in the feces compared to control groups. We did not observe any differences in the abundances of Bifidobacteria, Lactobacillus, Prevotella, Enterobacteriaceae, and Ruminococcus between the control and I3C-fed groups using PCR analysis in fecal samples.

## 4. Discussion

The results from the present study confirmed our hypothesis that dietary ingestion of I3C can modulate the gut microbiota. The major and novel observations from our study reside in observing an effect of dietary I3C on the gut microbiome and the association of such changes to tumor development. More specifically, ingesting I3C altered bacteria interaction, which appeared to be associated with promotion of tumor growth. These results support the notion that I3C and its metabolite DIM may protect against prostate cancer development indirectly through modulation of the gut microbiota.

The effect of dietary I3C on the gut microbiota appeared to be more subtle than we expected. There were no significant differences in microbial diversity, as indicated in alpha diversity analyses (Appendix A). I3C treatment did elicit a significant impact on the specific group of bacteria affiliated with phylum Deferribacteres, which was detected at the phylum level analysis as well as at species level. Additionally, non-hieratical analysis based on OTU level further identified changes in selected bacteria within the Firmicute phylum. However, there were no consistent changes to Firmicutes at species level, which were either down- or upregulated by I3C/DIM. Only *M. schaedleri*, which belongs to the Deferribacteres phylum, was enriched by I3C treatment. Perhaps the most dramatic changes caused by I3C/DIM are related to the modulation of bacterial interaction. Module 8 of the network interaction within the control group was found to be strongly associated with tumor growth, and altered by I3C intake. Hence, we speculated that in the case of prostate cancer prevention, specific interactions between the bacterial community may be more important than overall and absolute changes to bacteria population. It is possible that the gut microbiota adapts to its intestinal environment, such as through microbe-detected compounds changes, and forms a coordinated response without dramatically altering the bacterial composition or diversity. We inferred the gut microbiota may have sufficient plasticity to handle dietary changes. The causal effects of these changes on prostate carcinogenesis warrant further validation.

Correlation of specific bacteria to prostate cancer or the effect of I3C/DIM on prostate cancer may prove to be complex. The published literature related to gut microbiota and prostate cancer development is relatively limited and may lack validation. A recent study in 22 men with prostate cancer indicated that Gleason Score was positively associated with the phylum Deferribacteres, several Proteobacteria taxa, and Bacteroidetes. Gleason Score also was positively associated with *Clostridium* but inversely associated with *Blautia*, both from phylum Firmicutes [21]. We found Bacteroidetes phylum, through network analysis, was associated with tumor growth but not Proteobacteria. We did observe downregulation of Firmicutes via fecal PCR analysis. However, upregulation of Deferribacteres through metagenomic analysis would indicate that this phylum may be negatively associated with tumor growth. These findings may be a result of specie differences in mice and humans, or due to limited understanding of how bacteria are regulated by their environments. For example, specific bacteria may be more sensitive to environmental factors than others. Dietary I3C resulted in an enrichment of *M. schaedleri* in our study. It has been known that *M. schaedleri*, belonging to the Deferribacteres phylum, is an abundant habitant in the intestinal mucus layer of rodents but not humans. We suspect that this bacterium may be sensitive to I3C or its metabolite DIM. Interestingly, in Frugé’s article, Deferribacteres also responded to fruits and vegetables [21], and supported the notion that this phylum may be particularly sensitive to dietary changes. Additional studies are needed to delineate the importance of the changes observed in gut microbiota in prostate carcinogenesis, as well as the role and possible protective effects of diet-derived components.

*M. schaedleri*, as identified in our study as having responded to I3C/DIM-induced changes, reduced nitrate and modified the mucosal gene expression of its host. This bacterium also has specialized systems for scavenging oxygen and reactive oxygen species during inflammation. Additionally, it was reported to be positively correlated with serum leptin levels in a study of diet-modified obesity [34]. Hence, based on the known beneficial effect of *M. schaedleri* in different diseases, the I3C/DIM-induced increase of *M. schaedleri* may contribute to prostate cancer prevention and thus warrants further study.

Gut microbiota forms a complex community that is not only related to the number and the abundance of species, but also involves the interactions among microbial taxa. The network analysis of microbial co-occurrence patterns provided us with a new perspective to understand the structure of complex microbial communities, potential microbial interactions, and their ecological roles. Our data showed that while the topological properties of the global networks appear to be similar (Appendix A), the network composition was substantially differed between the control and I3C group (Figure 2). The networks from each of the respective diet groups had the same number of modules (8, Figure 2), which suggested that the interaction between the bacteria might be limited in the gut microbiota. However, few modules were functionally equivalent or shared similar node compositions. Furthermore, the topological roles of the nodes differed in these two networks. Each network had a distinct set of module hubs and connectors (Figure 3), which likely reflected habitat heterogeneity or trophic specialization under different dietary treatments.

As mentioned above, only one module (module 8) was significantly and positively correlated with tumor growth in the control group (*r* = 0.8, *p* = 0.009). All nodes in this module belong to the Bacteroidetes, Bacteroidales S24-7 family. This positive correlation between module eigengenes and tumor growth was disrupted by the I3C treatment. None of the other modules showed significant associations with tumor growth in the I3C group. S24-7 is an abundant member in the gastrointestinal tracts of animals. Different environmental conditions were able to alter the relative abundance of S24-7. For example, the abundance of S24-7 increased in diabetes-sensitive mice fed a high-fat diet [35]. Additionally, multiple studies suggested that some members of S24-7 are targeted by the innate immune system, implicating that it may be involved in host–microbiota interactions that influence gut function and host health [36,37]. Our findings provide further evidence that S24-7 may also be a pathological marker in prostate cancer and warrant further validation.

As mentioned above, the changes in cecal bacteria are variable even for bacteria from the same phylum, such as Firmicutes. Some bacteria from the Firmicutes phylum were upregulated and others were downregulated. In contrast to these changes, the result obtained from the PCR analysis of fecal samples showed that the Firmicutes in feces were determined to be downregulated. Although the cecal microbiota is quantitatively different from fecal microbiota, we consider this discrepancy may be due to the PCR analysis being skewed toward capturing changes in dominant species. Nonetheless, the significance of such changes/differences warrants further elucidation.

Given that cancer may modulate the gut microbiota [38], it is possible that inoculation of tumor cells can alter the gut microbiota. Therefore, it is possible that I3C/DIM may influence the changes resulting from tumor inoculation. Additional experiments are necessary to test this hypothesis. Moreover, it is probable that I3C-related changes in microbiota may not lead to tumor changes. Additional experiments that utilize fecal transplantation in gnotobiotic animals [39] may be used to determine whether microbiome changes elicited by I3C/DIM can affect tumor growth.

## 5. Conclusions

In summary, we found that the dietary intake of I3C significantly inhibited LNCaP cell tumor xenograft growth in mice. We identified that the chemo-preventive effects of diet-derived I3C/DIM may be associated with altered microbiota composition and a shift of microbial interactions in gut microbiota community. Additional work is warranted to elucidate the role of Bacteroidales S24-7, *M. schaedleri* in prostate carcinogenesis and the possible contributions of the cancer-reducing effects from dietary intake of I3C.

## Figures and Tables

**Figure 1 nutrients-11-00467-f001:**
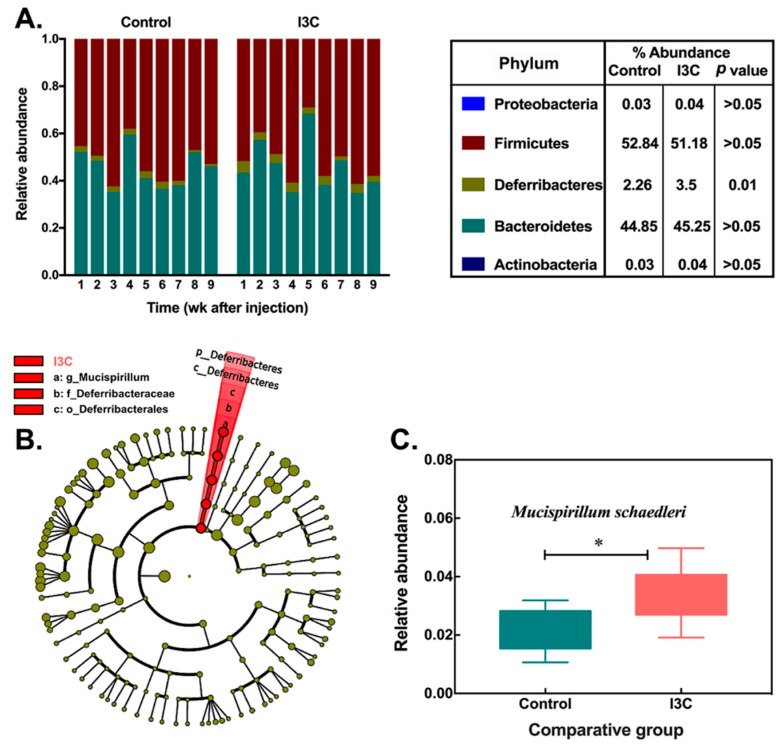
Alteration of cecal microbiota in control and indole-3-carbinol (I3C)-treated mice after LNCaP (ATCC CRL-1740) human prostate cancer cell injection. (**A**) the phylum-level microbial composition (bars represent relative abundance of samples from individual mice); (**B**) a cladogram representation based on operation taxonomic unit (out) table (the circles from inside to outside correspond to taxonomy from phyla to species, yellow and red circles represent non-significant and significant microbial clades, respectively); (**C**) the relative abundance of *Mucispirillum schaedleri* in control and I3C groups (* means *p* ≤ 0.05, Mean ± SD, *n* = 9).

**Figure 2 nutrients-11-00467-f002:**
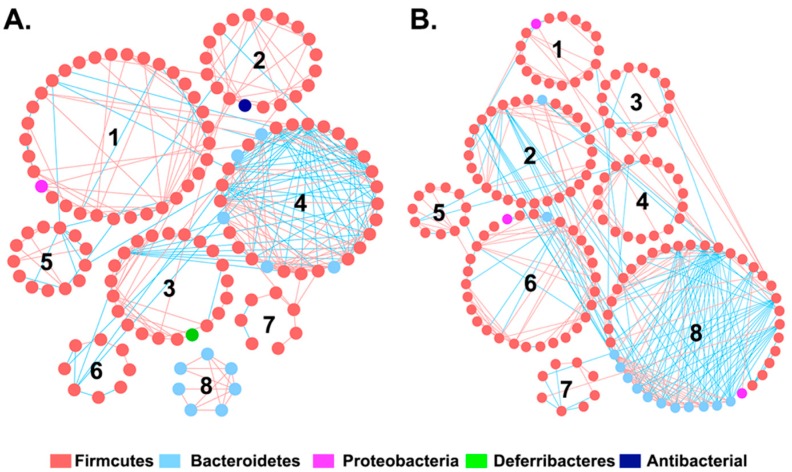
Intestinal microbiota network interactions under the (**A**) control and (**B**) I3C diet. Each node signified a species. Node colors indicate different phyla. A red line indicates a positive interaction between two individual nodes, suggesting a mutualism or cooperation, while a blue line indicates a negative interaction, suggesting predation or competition. The numbers 1 through 8 indicate number of modules in each network.

**Figure 3 nutrients-11-00467-f003:**
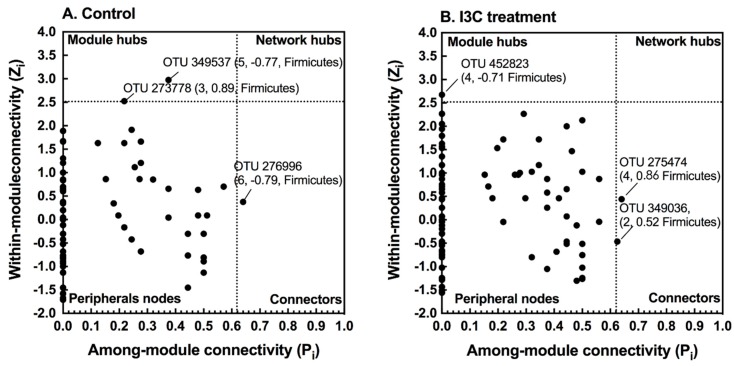
Z-P plot showing the distribution of OTUs based on their topological roles. I3C treatment greatly altered the network structure and topological roles of individual OTUs and key microbial populations. (**A**) Control; (**B**) I3C treatment. The topological role of each OTU was determined according to the scatter plot of within-module connectivity (Zi) and among-module connectivity (Pi). The module hubs and connectors are labeled with OTU numbers. In parentheses are the module number, module membership, and phylogenetic associations.

**Figure 4 nutrients-11-00467-f004:**
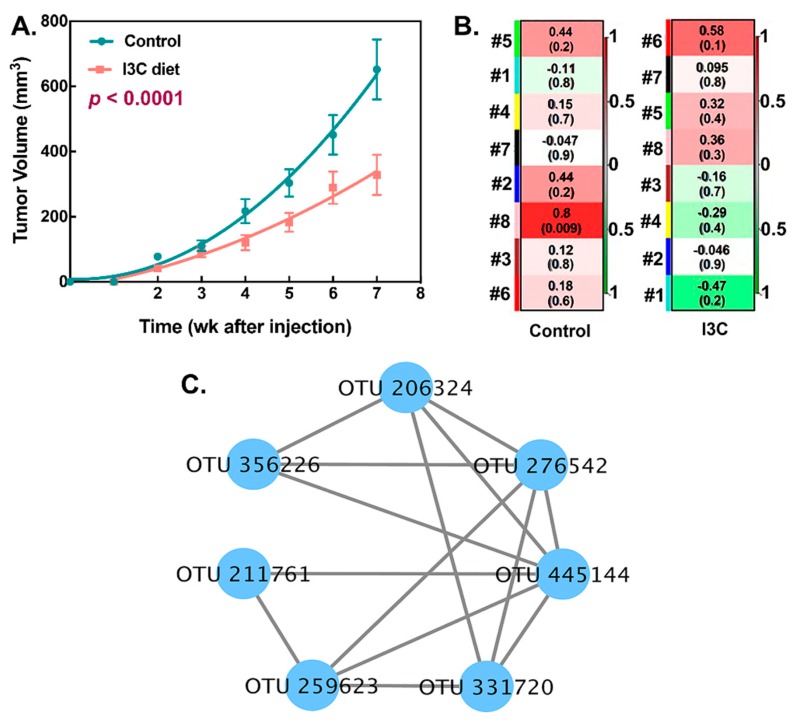
The relationship of network topology with environmental trait. (**A**) Effect of dietary I3C on tumor growth. (Mean tumor volume of tumors formed in control and I3C-treated mice during the 7-week treatment period after LNCaP human prostate cancer cell injection. The data obtained was analyzed by non-linear fitting and comparison of fits; two different curves were presented for each data set); (**B**) The correlation between module eigengene and tumor size (red color indicates a highly positive correlation and green color indicates a highly negative correlation. The numbers in each plot are the correlation coefficient (r) and significance (*p*) in parentheses); (**C**) The OTU composition of module 8 in the control group.

**Figure 5 nutrients-11-00467-f005:**
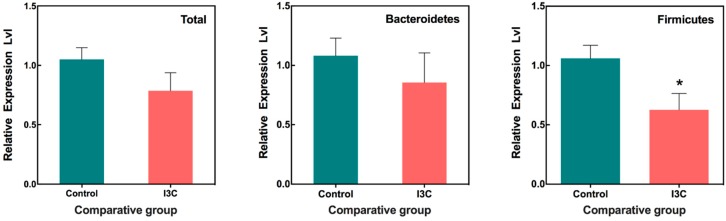
Alteration of fecal microbiota in mice fed with or without I3C. Bacterial DNA isolated from feces. Quantification of bacteria was performed with specific primers using qRT-PCR. Columns marked are significantly different from each other (*p* < 0.05, Mean ± SD, *n* = 9).

**Table 1 nutrients-11-00467-t001:** Sequence of real-time PCR primers.

Bacteria	Direction	Sequence (5′–3′)
Total bacteria	Forward	ACTCCTACGGGAGGCAG
	Reverse	GTATTACCGCGGCTGCTG
Bifidobacteria	Forward	TCGCGTCYGGTGTGAAAG
	Reverse	CCACATCCAGCRTCCAC
*Lactobacillus*	Forward	GAGGCAGCAGTAGGGAATCTTC
	Reverse	GGCCAGTTACTACCTCTATCCTTCTTC
*Prevotella*	Forward	TCCTACGGGAGGCAGCAGT
	Reverse	CAATCGGAGTTCTTCGTG
Enterobacteriaceae	Forward	CATTGACGTTACCCGCAGAAGAAGC
	Reverse	CTCTACGAGACTCAAGCTTGC
*Ruminococcus*	Forward	GGCGGCCTACTGGGCTTT
	Reverse	CCAGGTGGATAACTTATTGTGTTAA
Bacteroidetes	Forward	GGARCATGTGGTTTAATTCGATGAT
	Reverse	AGCTGACGACAACCATGCAG
Firmicutes	Forward	GGAGYATGTGGTTTAATTCGAAGCA
	Reverse	AGCTGACGACAACCATGCAC

**Table 2 nutrients-11-00467-t002:** Fourteen species-level OTUs with significant differences in relative abundance between the mice fed diet with or without I3C (*p* < 0.01). The numbers denote mean ± SD (percentage composition). Results are based on a Wilcoxon non-parametric *t*-test corrected for multiple hypothesis testing (LDA score log_10_ > 2.0).

Greengenes ID	Control	I3C	LDA log_10_ Score	Annotation
198238	0.29 ± 0.28	0.92 ± 0.59	3.5	Bacteria|Firmicutes|Clostridia|Clostridiales
311286	0.13 ± 0.01	0.05 ± 0.04	2.6	Bacteria|Firmicutes|Clostridia|Clostridiales|Lachnospiraceae
343923	0.02 ± 0.02	1.11 ± 2.15	3.8	Bacteria|Firmicutes|Clostridia|Clostridiales|Lachnospiraceae
195621	0.30 ± 0.31	1.05 ± 0.68	3.6	Bacteria|Firmicutes|Clostridia|Clostridiales
1510067	0.86 ± 0.37	0.44 ± 0.36	3.4	Bacteria|Firmicutes|Clostridia|Clostridiales
274697	0.15 ± 0.15	0.51 ± 0.34	3.3	Bacteria|Firmicutes|Clostridia|Clostridiales
747987	0.02 ± 0.06	0.20 ± 0.34	2.8	Bacteria|Firmicutes|Clostridia|Clostridiales
272757	0.17 ± 0.09	0.29 ± 0.13	2.8	Bacteria|Firmicutes|Clostridia|Clostridiales
332854	0.08 ± 0.05	0.03 ± 0.02	2.4	Bacteria|Firmicutes|Clostridia|Clostridiales
330219	0.03 ± 0.01	0.05 ± 0.02	2.1	Bacteria|Firmicutes|Clostridia|Clostridiales
314188	0.03 ± 0.01	0.05 ± 0.02	2.1	Bacteria|Firmicutes|Clostridia|Clostridiales
337494	0.01 ± 0.01	0.03 ± 0.02	2.1	Bacteria|Firmicutes|Clostridia|Clostridiales
1136443	2.26 ± 0.76	3.50 ± 0.93	3.8	Bacteria|Deferribacteres|Deferribacteres|Deferribacterales|Deferribacteraceae|*Mucispirillum*|*Schaedleri*

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
