# Peer review of "Inhibition of Tumor Growth by Dietary Indole-3-Carbinol in a Prostate Cancer Xenograft Model May Be Associated with Disrupted Gut Microbial Interactions"

_nutrients, 2019, doi:10.3390/nu11020467_

Reviewer 1 Report

The present manuscript is a research article about the role of indole-3-carbinol in preventing prostate cancer growth in a human prostate cancer xenograft mouse model, the authors hypothesized that the beneficial effect of I3C is due to its modulatory effect on gut microbial of mice, and they conduct a series of experiments including 16s rRNA gene-based sequencing and co-occurrence network analysis to verify their hypothesis. They concluded that there are some complex relationships among I3C ingestion, gut microbiota, and prostate cancer development and these may provide a novel insight into the mechanism for the chemopreventive effect of I3C on prostate cancer. These are significant findings. I have only a minor concern.

The authors only discussed the possible mechanisms of I3C may play on prostate cancer. Why not provide some simple experiments about I3C on prostate cancer cell lines cell proliferation, migration or invasion to confirm its preventive roles in vitro?

Author Response

Response to Reviewer 1 Comments

Point 1: The authors only discussed the possible mechanisms of I3C may play on prostate cancer. Why not provide some simple experiments about I3C on prostate cancer cell lines cell proliferation, migration or invasion to confirm its preventive roles in vitro?

Response: We thanks the reviewer for the suggestion. Actually, several groups, including ours, have reported in vitro effects of I3C on prostate cancer cells [9-10]. We reported I3C/DIM growth inhibitory effects on prostate cancer cell lines as well as tumor microenvironment [11]. We described/cited this works in the INTRODUCTION section. However, to make it clearer to the readers the following verbiage was added in the INTRODUCTION section, “It has been shown that I3C/DIM inhibit proliferation, migration and invasion of prostate cancer cells in vitro by targeting...” 

Reviewer 2 Report

This article describe the relationships among ingestion of  I3C and gut microbiota and prostatę cancer development.

The comments:

Abstract: please check the p value after ….tumor size…

Introduction: the Author should add more  information about gut microbiota

Materials and methods: the numer and consent of the ethics committee and the references genes used in RT-PCR technique are not supported

Discussion: I3C convert into DIM in acidic environment, so the Authors should take into consideration influence DIM instead I3C, 

Author Response

Response to Reviewer 2 Comments

Point 1: Abstract: please check the p value after …tumor size…

Response: Thanks for your suggestion. The p value for comparison of tumor size between control and I3C group was added into Abstract.

Point 2: Introduction: The author should add more information about gut microbiota.

Response: As suggestion by the reviewer we added more information about gut microbiota into the INTRODUCTION section.

Point 3: Materials and methods: the name and consent of the ethics committee and the

references genes used in RT-PCR technique are not supported

Response: As suggested by the reviewer, the name and consent of the ethics committee were added into the animal part of Materials and methods. Also, as suggested by the reviewer, the references genes for RT-PCR technique listed in the table in the support materials section was transferred into the Materials and methods.

Point 4: Discussion: I3C convert into DIM in acidic environment, so the authors should take into consideration influence DIM instead I3C

Response: We agree with the reviewer that DIM, as well as I3C may contribute to modulation of the microbiome as both compounds exist in the gut. We changed wording from I3C to I3C/DIM in the Discussion to reflect this.  

Reviewer 3 Report

In this manuscript (nutrients-429587), the authors demonstrated that dietary ingestion of indole-3-carbinol (I3C) significantly inhibited LNCaP xenograft growth via modulation of gut microbiota in mice.

It has been known that I3C has a potential to suppress the development of androgen-dependent human prostate carcinoma cells.  Therefore, it is difficult to evaluate the correlation between the alteration of microbiota and prostate cancer development by I3C in this experimental system. An additional experiment using the germ-free mouse and/or fecal transplantation is needed. 

Minor comments.

   The total number of mice used in this experiment and the number of mice of each group should be listed in the text.

Author Response

Response to Reviewer 3 Comments

Point 1: It has been known that I3C has a potential to suppress the development of androgen-dependent human prostate carcinoma cells.  Therefore, it is difficult to evaluate the correlation between the alteration of microbiota and prostate cancer development by I3C in this experimental system. An additional experiment using the germ-free mouse and/or fecal transplantation is needed.

Minor comments:

Response: We agree with the reviewer that androgen-dependent pathway is affect by I3C. However, we as well as others also shown that I3C may also modulate several other cellular pathways including xenobiotic metabolisms, inflammation pathway. Interaction of microbiome and these pathways may also occur and contribute to prostate cancer carcinogenesis. Therefore, to fully dissect contribution of microbiome on prostate carcinogenesis and the role of dietary I3C on these processes would involve multiple experiments and different animal models and beyond the scope of the present study. We do agree with the reviewer on the use of Germ-free mice/ fecal transplanted as a mean to further dissect the interaction between gut microbiome/I3C/prostate cancer. However, the role of gut microbiome on prostate cancer remain unclear and follow up study not only need to show role of microbiome and specific bacteria but also how dietary I3C can modulate gut microbiome and in term influence prostate cancer carcinogenesis. We felt these critical questions are complicated and more suitable for grant proposal and beyond the scope of a manuscript. We do believe that the results reported in our manuscript provide novel and exciting bases to further interrogate the utility of dietary components and prevention of prostate cancer through modulation of microbiome.

Point 2: The total number of mice used in this experiment and the number of mice of each group should be listed in the text.

Response: As suggested, the total number of mice in each group was added into the Materials and methods.

Round  2

Reviewer 1 Report

The manuscript has been largely improved after the author’s efforts. They have also explained my questions suitably.

Author Response

We thank the reviewer for his/her suggestions and comments.

Reviewer 3 Report

   It is hard to evaluate the preventive effects against prostate cancer by microbiota alteration in this system, but the microbiota modifying action by I3C is interesting simply.  This reviewer thinks that this is enough for this manuscript. The next article of this field is expected.

Author Response

We thank the reviewer for his/her suggestions and comments. We hope to follow up with our study soon